# Effectiveness of China’s Protected Areas in Mitigating Human Activity Pressure

**DOI:** 10.3390/ijerph19159335

**Published:** 2022-07-30

**Authors:** Jian Chen, Hong Shi, Xin Wang, Yiduo Zhang, Zihan Zhang

**Affiliations:** 1School of Tourism and Urban-Rural Planning, Chengdu University of Technology, Chengdu 610059, China; vgets@163.com (J.C.); beatrice0620@163.com (X.W.); 2School of Tourism and Historical Culture, Southwest Minzu University, Chengdu 610041, China; zhangyiduo@stu.swun.edu.cn (Y.Z.); zhangzihan@stu.swun.edu.cn (Z.Z.)

**Keywords:** protected areas, effectiveness, human activity pressure, China, SSP scenarios

## Abstract

Global protected areas are the key factor in maintaining biodiversity and ecosystem services. However, few studies use human activity pressure to assess the effectiveness of protected areas. This study constructed a human activity pressure index to assess the effectiveness of China’s protected areas, and predicted the change trend in 2050 under the SSP scenarios. The results are as follows: (1) From 2000 to 2020, the pressure of human activities in 75.15% of China’s protected areas is on the rise, accounting for 37.98% of the total area of the reserves. (2) China’s protected areas can relieve the pressure of human activities by 1.37%, and there are regional differences in the effectiveness. (3) Under the SSP scenarios, the protected areas can alleviate the effect of the pressure of the population well. These results can provide a systematic and scientific reference for the planning, construction, evaluation and management of global protected areas.

## 1. Introduction

The establishment of protected areas (PAs) is the main means to protect biodiversity and ecosystem services in the world. According to the International Union for Conservation of Nature (IUCN), the coverage of global protected natural areas increased from 14.7 percent in 2016 to 15.4 percent in 2021. Although the coverage of PAs is increasing, they are not completely immune to the impact of human activities [1,2].

Human activities can directly or indirectly damage biodiversity and ecosystem services, and can reduce the effectiveness of terrestrial PAs. Therefore, it is of great significance to prevent the unrestricted growth of human activities in or near natural PAs for the improvement of their conservation efficiency. China is one of the countries with large areas of PAs. They can be used as a representative to evaluate the conservation effectiveness of natural PAs. In the current studies, the traditional spatial comparison method is used to evaluate the conservation effectiveness of PAs in China, ignoring the problem of sample selectivity bias [1,3]. In addition, there are few studies that take anthropogenic pressure as an indicator to evaluate the conservation effectiveness [4]. There is also a general lack of research on the prediction of the impact of future anthropogenic pressure on the existing PAs in China. 

According to the bulletin on ecological and environmental conditions of China, more than 11,800 natural protected areas had already been established by 2019 in China, covering a total area of about 2 million km^2^, accounting for 18.0 percent of the country’s land area and 5.80 percent of the global protected areas. China is in a critical period of transition from focusing on the quantity of protected areas to paying attention to the role of protected areas. Therefore, it is of practical significance to select China as the research sample to study the ability of protected areas to relieve the pressure of human activities. The core question of this study is as follows: To what extent do the protected areas in China mitigate the pressures of human activities, and what will happen in the future? This paper has three aspects of innovation. In this study, a human activity pressure (HAP) index is constructed as an evaluation index. On the basis of solving the problem of sample selectivity bias by propensity score matching, this paper evaluates the effectiveness of China’s PAs in alleviating HAP. We also explore the impact of the pressure of a certain human activity on PAs in a future scenario. Firstly, this paper constructs the HAP from 2000 to 2020, coupled with multi-source data. Secondly, after the propensity score, the matching method was used to eliminate the sample selectivity bias, and the relative effectiveness index and panel model are used to evaluate the effectiveness of China’s PAs in alleviating the pressure of human activities from different levels. Finally, the propensity score matching method is used to evaluate the effectiveness of China’s PAs in alleviating population pressure and urban land pressure under the SSP scenarios. These research results can provide a scientific basis for global protected areas to relieve the pressure of human activities.

## 2. Literature Review

### 2.1. Human Pressure on PAs

In order to monitor the human pressure on a PA, we first need to obtain the data and develop the method to quantify human activity. The most common way is to monitor human pressure on PAs through field surveys. Harris et al. [5] recorded the direct human pressure (poaching) and indirect human pressure (pastoralism) on PAs by photographic survey, and explored their influences on the species richness, composition and activities of mammals. Meanwhile, Abukari and Mwalyosi [6] used questionnaire surveys to quantify the potential pressure on the PAs from the behavior of residents around the PAs. A field survey can obtain accurate and targeted data, but it costs a lot, and it is only suitable for small-scale research.

Large-scale studies usually construct human pressure data by coupling multi-source data which include basic geographic data, remote sensing images, and statistical data, etc. According to the number of pressure layers, human pressure data products can be divided into single pressure products and comprehensive pressure products.

A single pressure product is a data product that contains only one type of the HAP, that is, only one layer. It is usually obtained by basic surveying and mapping, the interpretation of remote sensing images, map vectorization, and the spatialization of statistical data. A single pressure product includes data of land use [7], transportation and accessibility [8], pollutants [9], population [10], animal husbandry [11], and invasive species [12], etc. Commonly used land use data are derived from remote sensing image interpretation, road and building profiles are derived from map vectorization, and demographic data generally comes from the spatialization of statistical data. Although NPP-VIIRS and DMSP/OLS, two kinds of nighttime light data that can be used as representatives of human activities, were difficult to use to complement each other in temporal resolution before 2020 [13], many studies have successfully integrated the two data recently [14,15], which has contributed to the extension of the time series of human pressure products.

The comprehensive pressure products are obtained by integrating multiple single-pressure layers. The modeling steps include selecting multiple single-pressure layers that represent human activities, assigning weight to each single-pressure layer, and selecting the integration manner of the layers. The selection of a single-pressure layer is mainly based on the availability of the layer, and the selection of a single-pressure layer to be integrated into a comprehensive pressure product should meet the requirements of the research on temporal resolution and spatial resolution, as well as the repeatability of measurement [16]. The weight distribution is mainly based on the research object, and it is carried out according to the influence degree of pressure on the PAs. The integration manner of the layers is relatively simple, and the method of direct summation is generally adopted. It is worth noting that several types of single-pressure layer selected for the comprehensive pressure product may be related to some extent, rather than mutually exclusive [17].

At present, there are two sets of dynamic integrated pressure products which are commonly used at the global scale. One is the Temporal Human Pressure Index (THPI) dataset [16], the other set is the dataset of the Global Terrestrial Human Footprint Map (GTHF) [18]. The THPI dataset combines three single-pressure layers of farmland, power facilities and population density to map the global comprehensive pressure products in 1990 and 2010, with a spatial resolution of about 10 km^2^. The GTHF dataset is a global dataset that covers eight single-pressure layers, including the built environment, population density, power facilities, farmland, pastures, roads, railways and navigable waterways, with an improved spatial resolution of 1 km^2^. It should be noted that the GTHF dataset contains data from 1993 and 2009, but only five of the single-pressure layers contain temporal information, and the pressure layers for roads, railways, and pastures are static [19].

### 2.2. A Review of Effectiveness Assessment of PAs

The purpose of monitoring human pressure on PAs is to assess the effectiveness of PAs in alleviating human pressure. At present, the research on reserve effectiveness assessment mainly takes habitat, ecosystem services and biodiversity as indicators. However, no matter what indicator is used to evaluate the effectiveness, the idea is still interlinked. Therefore, this section takes the assessment method as the clue to summarize the current research on the effectiveness assessment of PAs. Based on the ideas of Joppa and Pfaff [20], the methods of effectiveness assessment can be divided into the following three categories.

The first method is the direct assessment method, which only considers the status of the PAs itself and does not compare it with other areas. Jones et al. [21] used the human footprint dataset to quantify the pressure of human activities in the PAs, and they found a threshold where the pressure level is intense. They concluded that a third of the world’s PAs are under significant human pressure. Wu et al. [22] produced statistics on the proportion of terrestrial ecological areas, biodiversity priority areas and vegetation types in China’s PAs in order to evaluate the effectiveness of ecological diversity.

The second method is the spatial comparison method, that is, contrasting the PAs with all non-PAs or adjacent land (buffer zones). For example, Hellwig et al. [23] compared land cover changes in PAs, non-PAs and the one-kilometer buffer zone around PAs in Europe. Guette et al. [24] used the Digital Number (DN) value, namely the pixel brightness value, of remote sensing images of nighttime light data to quantify human activities. The trends of nighttime light DN values in global PAs, biodiversity hotspots and 25 km and 25–70 km buffer zones were explored. Duran et al. [25] revealed the overlap of the mining activities of four metals with global PAs and their 10 km buffer zones. Radeloff et al. [26] used housing growth data from 1940 to 2030 in the United States to quantify housing growth rates in wilderness areas, national parks, national forests, and their buffer zones. Qiu et al. [27] calculated three single-pressure indexes (population density, GDP, land use) and a comprehensive pressure index in 58 PAs and their 2 km buffer zones in Yunnan Province. Wang et al. [28] analyzed the forest protection situation of China’s national PAs and the whole country by using the dataset of the Global Forest Watch.

The third method is the time comparison method, which compares the protection effectiveness of the same place at different times, and generally compares the protection effectiveness before and after the establishment of the PAs. Zeng et al. [29] investigated the deforestation situation before and after the establishment of Wolong Nature Reserve, and found that deforestation increased after the establishment of Wolong Nature Reserve.

The direct evaluation method is the simplest evaluation method, but when using this method, the lack of reference will lead to the ambiguity of the evaluation standard, and it is difficult to draw a reasonable conclusion. The time comparison method can evaluate the effect of policy better; however, in the field of PAs, if there are too many reserves and they are established in different years, there will be confusion over the time period of the study sample. The spatial comparison method is the most commonly used evaluation method at present. By comparing the conservation results within the PAs with those outside the PAs, a more reasonable evaluation result can be obtained. However, this ignores the problem that the location of PAs is not randomly distributed, and is often affected by factors such as altitude, climate, accessibility, and so on [30,31]; it is difficult to distinguish whether the better protection efficiency in the PAs is due to the establishment of the PAs or the influence of the geographical background of the PAs. Therefore, some scholars began to use the matching method to solve the deviation problem of non-random positions [20]. At present, many studies have used the matching method to evaluate the effectiveness of PAs in mitigating forest loss [3,32,33,34], but little attention has been paid to the relief of the HAP on PAs. Geldmann et al. [35], for the first time, explored the mitigation of the pressure of human activities in the PAs through the matching method. In their research, several matching methods were compared; finally, propensity score matching (PSM) was chosen to eliminate the sample selectivity bias, and the effectiveness of the alleviation of the pressure of human activities in the PAs was reasonably evaluated, which proved the scientific nature of this evaluation method [36,37,38].

## 3. Materials and Methods

### 3.1. HAP Index

Human activity pressure data products are divided into single-pressure products and comprehensive pressure products. At present, comprehensive pressure evaluation is the mainstream evaluation method, among which Venter et al. [19] constructed the human footprint method, which is the most representative, and estimated the global human activity pressure. Therefore, this paper selects four kinds of single HAP—namely urban land, farmland, population and power facilities—to construct a comprehensive HAP index.

The growth of roads in the PAs cannot be ignored, but it is difficult to obtain real-time updated road data. For example, due to the difficulty in obtaining data, Venter et al. [18] adopted static road data in their research; as such, the road pressure is omitted in this paper. The scoring rules for each individual pressure index are as follows. The direct assignment method is used to calculate the pressure of urban land and agricultural land. Here, the score for urban land is 10, while the score for agricultural land is 7.

The modeling of population pressure adopts the formula method. It is assumed that the population pressure on the natural system increases logarithmically with the increase of the population density, and that it can reach saturation after a certain point; that is to say, the population pressure remains unchanged after the population density increases to a certain degree. For this hypothesis, when the population density is greater than 1000 people/km^2^, the score of the population pressure is 10; when the population density is lower than 1000 people/km^2^, the score of population pressure is calculated by the following formula: (1)$$pps=3.333\times \log_{10}(pd+1)$$
where pps is the population pressure score, and pd represents the population density.

The local technology development and the use of fossil fuels can be estimated well by the situation of power facilities (Sanderson et al. [17]). In this paper, taxonomy is used for the value assignment of power facilities. The specific approach is as follows: the DN value of night light data in 2000 is divided into 0~10 by the quantile method after preprocessing. In order to make the data of different years comparable, the DN value of the night light data in 2005, 2010, 2015 and 2020 is also divided into 0~10 according to the same method.

Finally, four single scores are added together to obtain a comprehensive HAP index product with a range of 0–30 and a spatial resolution of 1 km × 1 km. The construction of the HAP index is shown in Equation (2).
(2)$$s_{all}=s_{urban}+s_{crop}+s_{pop}+s_{elec}$$
where $s_{all}$ is the pressure score of comprehensive human activities, $s_{urban}$ represents the pressure score of urban land, $s_{crop}$ represents the pressure score of agricultural land, $s_{pop}$ is the pressure score of the population, and $s_{elec}$ is the pressure score of electric power facilities.

### 3.2. Propensity Score Matching Method

In order to eliminate the selection bias of the samples, this paper uses the propensity score matching method. The matching method can be used to select a treatment group and a control group with similar initial conditions. The idea is that, for the individual *i* in the treatment group, we can find an individual *j* in the control group, and make the values of the observation variables of the two as similar as possible ($X_{i}\approx X_{j}$). Generally speaking, the observed variable is a multi-dimensional vector. The propensity score matching method takes the propensity score as the distance function, and matches according to the rules of the nearest neighbor or whole matching. The propensity score of individual *i* is defined as the probability of individual *i* entering the treatment group under the given observation variables, and its calculation method is expressed as follows:(3)$$p(X_{i})=\mathrm{pr}(PA_{i}=1|X_{i})=F(h(X_{i}))$$
where $p(X_{i})$ represents the propensity score of individual *i*, $PA_{i}$ is the dummy variable, $PA_{i}=1$ is the treatment group, and $PA_{i}=0$ is the control group. $X_{i}$ is a series of observed variables, h(.) is a linear function, and F(.) is a logical function.

### 3.3. The Panel Model

This paper uses the panel model to study the effectiveness of China’s protected areas at mitigating human activity pressure. The formula of the model is as follows:(4)$$Y_{it}=a_{0}+a_{1}PAS_{it}+a_{2}X_{it}+\varepsilon_{it}$$
where $Y_{it}$ represents the HAP at random point *i* in year *t*, $PAS_{it}$ is the individual dummy variable, $PAS_{it}=1$ is the treatment group, and $PAS_{it}=0$ is the control group. $X_{it}$ is a series of control variables, and $\varepsilon_{it}$ is the random error term. Because the virtual PAS does not change with time, a fixed effect model cannot be used for the panel model; as such, the random effect model is adopted.

### 3.4. Data Source and Processing

According to different purposes, the data used in this paper is divided into three types: the data of PAs, data used to construct the HAP index, and other data (Table 1). In the process of preprocessing, WGS1984 (National Geospatial-Intelligence Agency, Springfield, VA, USA) cylindrical equal-area projection is used uniformly for all of the spatial data.

## 4. Results

### 4.1. Spatial Distribution of the HAP Index in China and Its PAs

#### 4.1.1. Spatial Distribution of the HAP Index in China

The spatial distribution of the HAP index constructed by the above method is shown in Figure 1. It can be seen that from 2000 to 2020, with the acceleration of China’s urbanization process, the pressure of human activity starts to spread from the urban center to the surrounding areas, and most of the pixels of which the HAP index values are within the range [13.65, 17.05] enter into a higher range. In total, 80.78% of the country’s land is subjected to different degrees of the HAP, and the proportion of protected land of which the HAP index is greater than 0 is 45.96% of the total protected land, which is lower than the proportion of the country’s land. 

#### 4.1.2. Spatial Distribution of the HAP Index in PAs in China

In this paper, trend analysis is used to analyze the change trend of the HAP index. Specifically, the least square method is used to calculate the slope of the HAP index over five years in order to represent the change trend of the HAP index during 2000–2020. As can be seen from Figure 2a, the areas where the change trend of the HAP remains unchanged or decreases are mainly in some central urban areas, the Qinghai–Tibet Plateau, Xinjiang, Inner Mongolia, Sichuan–Chongqing, and other areas. The HAP index in most of China’s land shows an upward trend, and this trend also appears in PAs (Figure 2b).

The land area showing an upward trend account for 64.71% of the total area, while the reserve area showing an upward trend account for 37.98%, and the number of reserve areas showing an upward trend account for 75.15%. On the whole, the change range of the HAP index is not obvious, and the slope concentrates around value 0.

### 4.2. Balance Test of the Propensity Score Matching Method

In this paper, altitude, slope, precipitation, temperature, the distance to a road, and the distance to urban land are selected as observation variables. First, random points are created in the PAs established before 2000 with an interval of no less than 1 km, and 20,297 random points are obtained. Because the resolution of the HAP index product constructed in this paper is 1 km, if the interval is less than 1 km, there may be more than one random point in a pixel, which does not meet the sampling requirements. If the interval is greater than 1 km, the sample size will be insufficient. Second, 180,000 random points are created outside the PAs. In order to avoid the spillover effect of the PAs affecting the matching results, random points are not generated within the 10 km buffer zone of the PAs [33]. Finally, the pixel values of the HAP and observed variables in 2000 are extracted to the random points, and the nearest-neighbor caliper matching method is used for matching. According to the recommendations of Rosenbaum and Rubin [39], a quarter of the sample standard deviation of the propensity score is selected as the caliper size. After the matching process, the random points inside and outside the PAs are 19,724 and 173,347, respectively.

The balance test results of the propensity score matching method are shown in Table 2. As can be seen from the change of the T statistic, at the level of 0.05, the characteristic variables of the two groups of random points no longer have significant differences after matching. Rosenbaum and Rubin [39] pointed out that if the absolute value of standard deviation after matching is less than 20%, it means that the matching is effective. In this study, the absolute values of the standard deviations are all less than 5%. This shows that the characteristic variables and matching method selected in this paper are reasonable.

### 4.3. Assessment of China’s PAs regarding the Alleviation of the Pressure of Human Activities

The panel model was used to evaluate the effectiveness of PAs in various geographical zones regarding the alleviation of the pressure of human activities. In order to construct the panel model, the sample data of the HAP and control variables in and outside the PAs from 2000 to 2020 should be obtained. In this paper, the elevation, slope, average annual precipitation and average annual temperature are selected as the control variables. Firstly, the dummy variable, PAS, in this paper does not change with time, such that the random points in 2005, 2010, 2015 and 2020 are the same as those in 2000 after matching. Secondly, these random points are used to extract the pixel values of the HAP index and the control variables in the five years. Finally, this paper chooses the random effects model for estimation.

The results of the panel model are shown in Table 3. As shown in the second column, at the national level, China’s PAs can relieve 1.37% of the pressure of human activities. Moreover, there are regional differences in the ability of PAs to relieve the pressure of human activities. The reserves in Northeast, East and Central South China have the most significant ability to relieve the pressure of human activities, with coefficients of −0.339, −0.328 and −0.199, respectively. The PAs in Southwest and Northwest China do not relieve the pressure of human activities but increase the pressure of human activities. This may be due to the influence of policies, such as the policy for the development of the western region, which led to the rapid development of the western region from 2000 to 2020.

### 4.4. Assessment of the HAP Mitigation of PAs in China under the SSP Scenarios

#### 4.4.1. Population Pressure Mitigation of PAs in China under the SSP Scenarios

The relative effectiveness index is used to evaluate the population pressure relief efficiency of PAs in China under the SSP scenarios. The relative effectiveness indexes of China’s PAs in the alleviation of the three population pressures under the five scenarios are obtained (Table 4). It can be seen that under the five SSP scenarios, the average of the relative effectiveness indexes of China’s PAs in alleviating the pressure of the population is negative, showing good pressure relief efficiency. When alleviating the pressure of the population, the effectiveness of the PAs under the five SSP scenarios ranked as SSP4 > SSP1 ≈ SSP5 > SSP2 > SSP3. 

In order to directly reflect the increase of the population pressure index in and outside the PAs, this paper calculates the average increase of the population pressure index after matching. Figure 3 shows the average growth amount of the population pressure index in China and its six major regions under the SSP scenarios. Because the population of China is gradually decreasing under the SSP scenarios, the decreasing trend is also reflected in the population pressure index. The decreasing values of the population pressure index under the five SSP scenarios are different to some extent, but the overall pattern is similar. Except for Northeast China, the average decrease of the population pressure index in the PAs is higher than that outside the PAs.

#### 4.4.2. Urban Land Pressure Mitigation by PAs in China under the SSP Scenarios

Similarly, the relative effectiveness index is also used to evaluate the effectiveness of China’s PAs in alleviating urban land pressure under the SSP scenarios. First, after the matching process, the random points outside the PAs are extracted, and their average growth of urban land pressure under the five scenarios is taken as a benchmark. By subtracting their respective benchmarks from the average growth of urban land pressure in the PAs under the five scenarios, the relative effectiveness indexes of China’s PAs in alleviating urban land pressure under the five scenarios are finally obtained (Table 5). It can be seen that under the five SSP scenarios, because most PAs are far away from urban land, they are almost not affected by urban land pressure. Specifically, the proportion of PAs with a negative relative effectiveness index is more than 90%.

Although the number of PAs with poor performance is small (about 5%), the average value of the relative effectiveness index is positive, indicating that the PAs with poor performance contribute more to the value of the relative effectiveness index. That is to say, from 2020 to 2050, the alleviation efficiency of PAs against the pressure of urban land use is not high. When alleviating the urban land pressure, the effectiveness of the PAs under the five SSPs scenarios was ranked as SSP3 > SSP2 > SSP4 > SSP5 > SSP1.

In order to directly reflect the growth of urban land pressure index in and outside the PAs, this paper calculates the average growth of the urban land pressure index after matching. Figure 4 shows the average growth amount of the urban land pressure index in China and its six regions under the SSP scenarios. For example, under the SSP2 scenario, there is almost no increase in urban land pressure within the PAs, while under the SSP3 scenario, the increase in urban land pressure within the PAs in East China even exceeds that outside the PAs.

## 5. Discussion

### 5.1. The Pressure of Human Activities in 75.15% of China’s PAs Shows an Upward Trend

In this study, the HAP in about three quarters of China’s PAs showed an increasing trend in the last 20 years, which indicates that human activities in and around the PAs are increasing. These human activities include the development of mines, roads, hydropower stations, power transmission lines and tourist attractions in the PAs. Many countries in tropical regions even rely on PAs to develop tourism and promote national economic development by obtaining foreign exchange income from tourism [40]. Qiu et al. [27] also confirmed that the population and economic activities in 58 PAs in Yunnan Province were gradually increasing. The pressure of human activities in 75.15% of China’s PAs shows an upward trend. This conclusion also indicates that although China has gradually formed a natural reserve protection system with national parks as the main body, PAs as the basis, and various kinds of natural parks as the supplement, it is still necessary to establish a real-time monitoring system based on big data in order to monitor the HAP in Pas, and to avoid irreversible impacts of human activities on habitats, ecosystem services or biodiversity in PAs. Therefore, countries and regions need to conduct the dynamic assessment and management of global PAs, which can be considered as part of the social governance system [41].

### 5.2. Regional Differences in China’s PAs in Alleviating the Pressure of Human Activities

There are regional differences in the ability of PAs to alleviate the pressure of human activities. In the past two decades, the PAs in Northeast, East and Central China have effectively alleviated the pressure of human activities; however, the PAs in Southwest and Northwest China have not relieved the pressure of human activities but increased the pressure of human activities. This may be due to the influence of policies, such as the “Great Western Development Policy”, which led to the rapid development of the western region from 2000 to 2020.

For example, by 2015, 81 counties (cities and districts) in Guizhou were connected at high speed, making Guizhou the ninth province in China and the first province in the west to realize the connection between counties and counties. These measures have effectively changed the accessibility and investment and financing environment of Guizhou. Compared with other western provinces, the movement of the human flow has increased. In addition, the conservation of PAs is not uniform across China’s provinces, which may also contribute to the differences in the relief of the HAP in PAs [42].

## 6. Conclusions 

Taking HAP as an indicator, this paper evaluates the effectiveness of China’s PAs in alleviating HAP, and explores the alleviating situation of specific HAPs in China’s PAs under the SSP scenarios; we drew the following conclusions:(1)From 2000 to 2020, the pressure of human activities began to spread from the urban center to the surrounding areas. In total, 80.78% of China’s land is under pressure from human activities to varying degrees, while only 45.96 % of the protected land has a HAP index greater than 0. The land area with a rising trend of the HAP index accounts for 64.71% of the total area, and 75.15% of the reserves show an upward trend in the HAP index, but their area accounts for only 37.98% of the total area of PAs.(2)PAs in China can relieve the pressure of human activities by 1.37%, and there are regional differences in the ability of PAs to alleviate the pressure of human activities. The reserves in Northeast, East and Central China have the most significant effects on relieving the pressure of human activities, with coefficients of −0.339, −0.328 and −0.199, respectively. Meanwhile, PAs in Southwest and Northwest China are increasing the pressure of human activity.(3)Under the five SSP scenarios, the urban land pressure index shows an overall increasing trend, and the increase is mainly concentrated in Eastern and Central China, but there is no similar pattern under the five SSP scenarios. For example, under the SSP3 scenario, the average increase of urban land pressure in the Eastern China PAs exceeds that outside the PAs.

## 7. Implications

In order to prevent the disorderly increase of the HAP in PAs, China should build a long-term supervision system for PAs in the future. The recommended value and maximum allowable value of the HAP should be set nationwide, and each major region, province, city and county should formulate different strategies to limit growth of the HAP in PAs according to the actual situation, such as development needs and ecological fragility.

The impact of the HAP on PAs in China under the SSPs scenario requires the comprehensive consideration of the complex socioeconomic situation in China. Urban expansion and the migration of the rural population to cities will change the relationship between humanity and the natural system. Urban expansion directly or indirectly occupies ecological space and destroys part of the natural ecosystem [43]. However, the migration of the rural population to cities may change the original livelihood methods of rural residents (poaching, deforestation, etc.). A reasonably organized population can make more intensive use of resources, and can finally enable the world to find ways to alleviate ecological environmental degradation.

There is room for further research to assess the effectiveness of PAs in alleviating the pressure of human activity. First of all, the quality of the HAP products needs to be improved. In terms of data, remote sensing and big data can be used as representatives of human activities [4]. However, there are certain differences in data sources, resolution and accuracy, such that it is difficult to select data that can fully represent human activities.

Secondly, in terms of studies evaluating the effectiveness of PAs, this paper presents a new potential approach: the pressure of human activities is used as an indicator [35,44] or the state of natural ecosystems is used as an indicator [16,22,32]. However, the interaction between human activities and nature is a complex process, and unilateral assessment has certain limitations. Future research should focus on the interaction process between human activities and nature, so as to provide more reasonable guidance for the planning, construction, evaluation and management of PAs.

## Figures and Tables

**Figure 1 ijerph-19-09335-f001:**
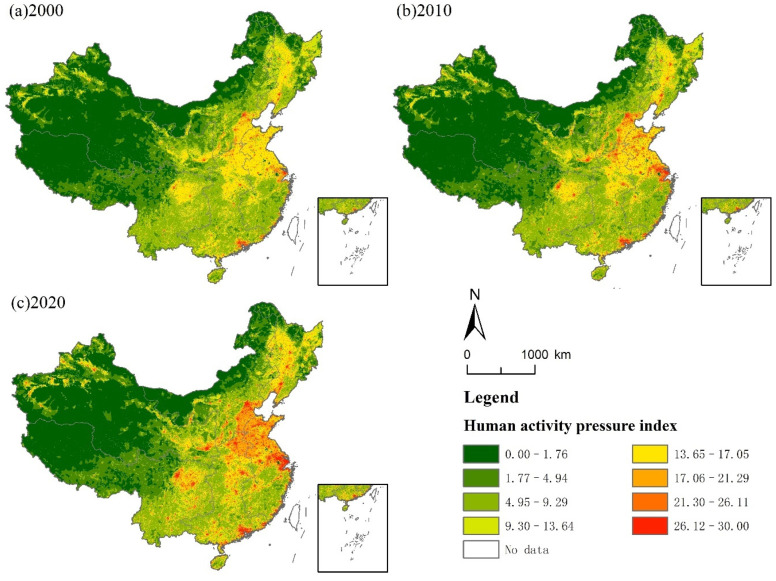
Spatial distribution of the HAP index in China.

**Figure 2 ijerph-19-09335-f002:**
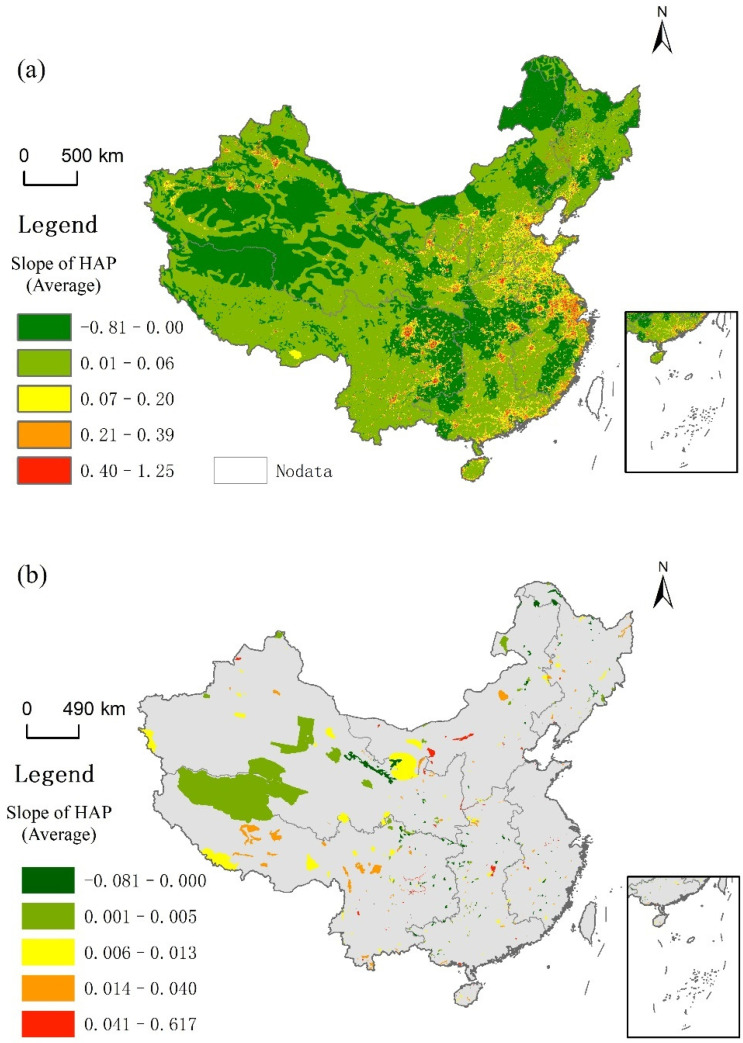
Spatial distribution of the HAP index in China and its PAs ((**a**)slope of HAP in China, (**b**) slope of HAP of protected areas in China).

**Figure 3 ijerph-19-09335-f003:**
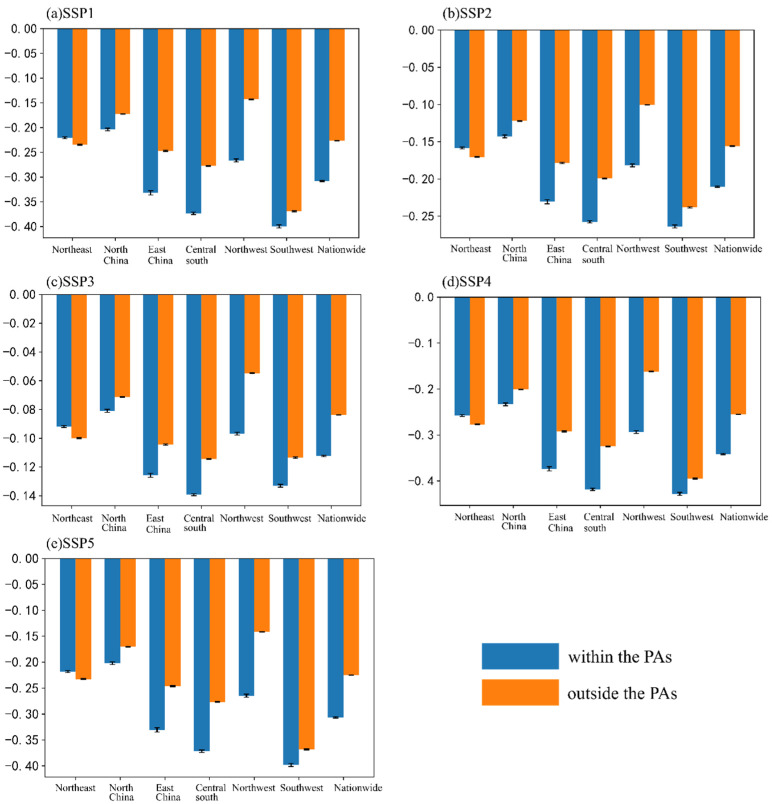
Average increase of the population pressure index in 2020–2050 under the SSP scenarios.

**Figure 4 ijerph-19-09335-f004:**
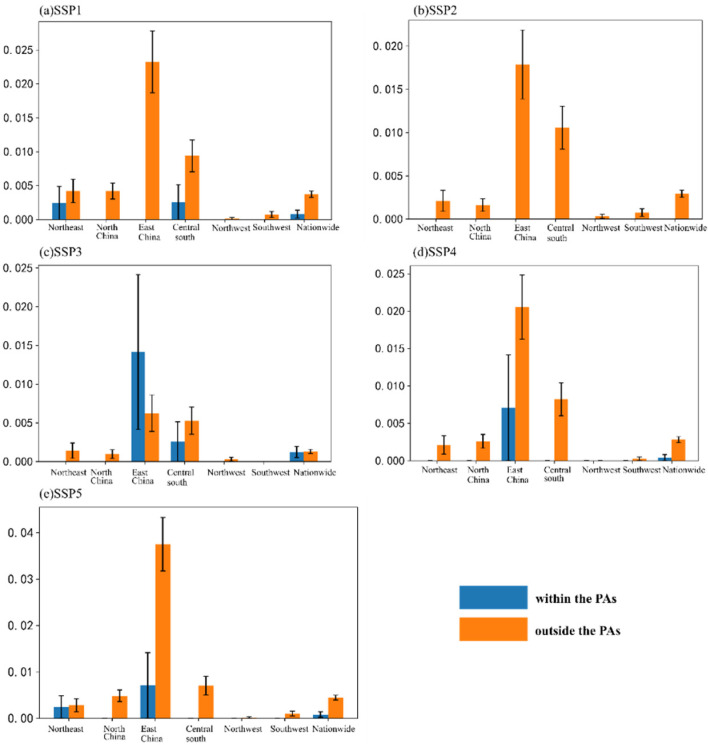
Average increase of the urban land pressure index in 2020–2050 under the SSP scenarios.

**Table 1 ijerph-19-09335-t001:** The dataset used in this article.

Data Category	Nature	Year	Data Source
**Data of PAs**			
Boundary data of PAs in China	Vector data	1957–2012	Geographic Information Database of China Nature Reserve Specimen Resource Sharing Platform http://www.papc.cn/html/folder/946895-1.htm (accessed on 20 May 2019)
List of National Nature Reserves (2017)	Text data	2017	Ministry of Ecology and Environment of the People’s Republic of China. http://www.mee.gov.cn/ (accessed on 20 May 2019)
**Data used to construct HAP index**			
Land cover data	Raster data (300 m)	2000–2019	European Space Agency, http://maps.elie.ucl.ac.be/CCI/viewer/index.php (accessed on 20 May 2019)
Future urban land use data	Raster data (300 m)	2020–2050	https://doi.pangaea.de/10.1594/PANGAEA.905890 (accessed on 4 July 2020)
Population density data	Raster data (1 km)	2000–2020	NASA Center for Socio-Economic Data and Applications. https://sedac.ciesin.columbia.edu/data/set/gpw-v4-population-density-rev11 (accessed on 4 July 2020)
Future population data	Raster data (1 km)	2020–2050	NASA Center for Socio-Economic Data and Applications. https://sedac.ciesin.columbia.edu/data/set/popdynamics-1-km-downscaled-pop-base-year-projection-ssp-2000-2100-rev01 (accessed on 4 July 2020)
Complementary DMSP and VIIRS night light data	Raster data (1 km)	2000–2018	https://figshare.com/articles/dataset/Harmonization_of_DMSP_and_VIIRS_nighttime_light_data_from_1992-2018_at_the_global_scale/9828827/2 (accessed on 4 July 2020)
**Other data**			
Temperature	Text data	2000–2020	NOAA National Environmental Information Center Database https://www.ncei.noaa.gov/data/global-summary-of-the-day/archive/ (accessed on 4 July 2020)
Precipitation	Text data	2000–2020	NOAA National Environmental Information Center Database https://www.ncei.noaa.gov/data/global-summary-of-the-day/archive/ (accessed on 4 July 2020)
Elevation	Raster data (1 km)		Food and Agriculture Organization of the United Nations http://www.fao.org/soils-portal/soil-survey/soil-maps-and-databases/harmonized-world-soil-database-v12/zh/ (accessed on 4 July 2020)
Data set of major roads across the country	Vector data	2000	Geographic Data Platform of Peking University https://geodata.pku.edu.cn/index.php?c=content&a=show&id=1399 (accessed on 4 July 2020)
National road data sets	Vector data	2018	Geographic Data Platform of Peking University. https://geodata.pku.edu.cn/index.php?c=content&a=show&id=713 (accessed on 4 July 2020)
Land area data	Raster data (1 km)	2010	NASA Center for Socio-Economic Data and Applications. https://sedac.ciesin.columbia.edu/data/set/gpw-v4-land-water-area-rev11 (accessed on 4 July 2020)

**Table 2 ijerph-19-09335-t002:** Balance test results of the propensity score matching method.

Variable		Mean Value		Standard Deviation (%)	Reduction in the Standard Deviation (%)	*t*-Test	
		The Treatment Group	The Control Group			*t* Statistic	Associated Probability of the *t*-Test
lnpecp	Before the match	−2.7261	−3.1	42.1		52.90	0.000
	After the match	−2.7261	−2.7284	0.3	99.4	0.26	0.796
lntemp	Before the match	3.8746	3.8061	26.9		34.26	0.000
	After the match	3.8746	3.8794	−1.9	92.9	−1.91	0.056
lnslope	Before the match	0.7148	0.21309	27.6		37.80	0.000
	After the match	0.7148	0.70359	0.6	97.8	0.63	0.528
lnelev	Before the match	6.7748	6.7164	4.0		5.14	0.000
	After the match	6.7748	6.7708	0.3	93.2	0.28	0.781
lntourban	Before the match	10.333	10.392	−5.5		−6.58	0.000
	After the match	10.333	10.321	1.1	80.5	1.08	0.279
lntoroad	Before the match	9.9819	10.026	−3.7		−4.56	0.000
	After the match	9.9819	9.9951	−1.1	70.4	−1.11	0.265
lnlandcover	Before the match	0.80104	0.8439	−7.1		−8.70	0.000
	After the match	0.80104	0.80166	−0.1	98.6	−0.11	0.913

Note: lnpecp represents precipitation, lntemp represents temperature, lnslope represents the slope, lnelev represents elevation, lntourban represents the distance to urban land, lntoroad represents the distance to a road, and lnlandcover represents the landcover type.

**Table 3 ijerph-19-09335-t003:** Panel model estimation results.

Variable	(1)	(2)	(3)	(4)	(5)	(6)	(7)
	Nationwide	Northeast China	North China	East China	Central South Region	Northwest China	Southwest China
PAS	−0.0137 *	−0.339 ***	−0.0388 *	−0.328 ***	−0.199 ***	0.0465 **	0.0482 ***
	(0.0070)	(0.0158)	(0.0206)	(0.0106)	(0.0062)	(0.0202)	(0.0112)
lnelev	−0.491 ***	−0.279 ***	−0.616 ***	−0.140 ***	−0.141 ***	−0.797 ***	−0.911 ***
	(0.0018)	(0.0080)	(0.0056)	(0.0032)	(0.0029)	(0.0091)	(0.0045)
lnslope	0.112 ***	−0.0825 ***	0.216 ***	−0.0651 ***	−0.0873 ***	0.109 ***	0.0832 ***
	(0.0016)	(0.0044)	(0.0043)	(0.0025)	(0.0020)	(0.0039)	(0.0033)
lnpecp	0.0635 ***	0.00874 ***	0.0460 ***	−0.00984 ***	−0.0240 ***	0.102 ***	0.0202 ***
	(0.0004)	(0.0014)	(0.0009)	(0.0016)	(0.0009)	(0.0007)	(0.0011)
lntemp	0.319 ***	0.140 ***	0.0735 ***	0.675 ***	0.222 ***	0.533 ***	0.366 ***
	(0.0018)	(0.0074)	(0.0041)	(0.0088)	(0.0048)	(0.0058)	(0.0023)
_cons	3.577 ***	2.952 ***	5.138 ***	0.202 ***	2.072 ***	5.138 ***	6.736 ***
	(0.0142)	(0.0549)	(0.0414)	(0.0401)	(0.0261)	(0.0735)	(0.0358)
*N*	807292	79403	138076	82089	111441	195871	200343
*R* ^2^	0.4122	0.2150	0.2584	0.4839	0.4793	0.2188	0.5306

Note: PAS is a dummy variable, lnelev represents elevation, lnslope represents the slope, lnpecp represents precipitation, lntemp represents temperature. The numbers in the brackets indicate the standard error, * represents significance at the level of 0.1, ** represents significance at the level of 0.05, and *** represents significance at the level of 0.01.

**Table 4 ijerph-19-09335-t004:** Relative effectiveness index of PAs in China in alleviating population pressure under the SSP scenarios.

SSPs Scenario	Number of PAs	Index (Negative, %)	Index (Positive, %)	Relative Validity (Average)
SSP1	670	74.03	25.97	−0.076
SSP2	670	76.72	23.28	−0.054
SSP3	670	79.55	20.46	−0.031
SSP4	670	76.12	23.88	−0.087
SSP5	670	74.03	25.97	−0.076

**Table 5 ijerph-19-09335-t005:** Relative effectiveness index of PAs in China in alleviating urban land pressure under SSP scenarios.

SSPs Scenario	Number of PAs	Index (Negative, %)	Index (Positive, %)	Relative Validity (Average)
SSP1	670	94.63	5.37	0.032
SSP2	670	95.37	4.63	0.022
SSP3	670	95.67	4.33	0.016
SSP4	670	94.93	5.07	0.028
SSP5	670	93.88	6.12	0.031

## Data Availability

Some or all data and models that support the findings of this study are available from the corresponding author upon reasonable request.

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
