# Peer review of "Effectiveness of China’s Protected Areas in Mitigating Human Activity Pressure"

_ijerph, 2022, doi:10.3390/ijerph19159335_

Round 1
Reviewer 1 Report
Very interesting manuscript. Manuscript is very well written and structured. Despite it requires some minors English spell-check, it is an interesting paper. I have some comments and suggestions for improving, especially the potential impact of this research for international environmental planning (and not only for China´s case), because this aspect is completely missing in this paper.
I consider that this work could be benefit from the following:
- Abstract needs improvements: first, problem statement should be inserted within the international context and not only in China context. Second, methodology should be precised and described briefly. Third, impact/significance of results should consider which could be the potential impact to the international state of the art and not only to China context.
- Phrase in line 37-38: The lack of research on predicting the impact of future anthropogenic pressure occurs only in China? or, is it occurs at international level also? Please precise and clarify.
- Lines 39-41: Why China is selected as case study? Please extend your arguments
- Lines 47-48: The international relevance of this contribution should be highlighted and mentioned. Also, a research question should be formulated at this section. Also, hypothesis mentioned in method section should be inserted in introduction section.
- Lines 166-167: Why this kind of HAP are selected? This should be explained more in depth in this paragraph.
- Figures 1 and 2 should be improved in terms of quality.
- Conclusions should discussion about the potential implications of result to the international context, especially in terms of validity of the method applied. A new paragraph could be inserted in line 424, or in line 432.
Author Response
- Abstract needs improvements: first, problem statement should be inserted within the international context and not only in China context. Second, methodology should be precised and described briefly. Third, impact/significance of results should consider which could be the potential impact to the international state of the art and not only to China context.
Response: Line 10-19 on page 1. Thanks for the comments. We accepted the suggestions, and revised the abstract.
“Global protected areas is the key factor in maintaining biodiversity and ecosystem services. However, few studies use human activity pressure to assess the effectiveness of protected areas. This study constructed the human activity pressure index to assess the effectiveness of China’s protected areas, and predicted the change trend in 2050 under the SSPs scenario. The results show that :(1) From 2000 to 2020, pressure of human activities in 75.15% of China's protect-ed areas are on the rise, accounting for 37.98% of the total area of the reserves. (2) China's protected areas can relieve the pressure of human activities by 1.37%, and there are regional differences in the effectiveness. (3) Under the SSPs scenario, the protected areas can alleviate effect on the pressure of population well. These results can provide a systematic and scientific reference for the planning, construction, evaluation and management of global protected areas.”
- Phrase in line 37-38: The lack of research on predicting the impact of future anthropogenic pressure occurs only in China? or, is it occurs at international level also? Please precise and clarify.
Response: Line 37-38 on page 1. Thanks for the comments. We accepted the suggestions, and revised it.
“There is also a general lack of research on predicting the impact of future anthropogenic pressure on the existing PAs in China”
- Lines 39-41: Why China is selected as case study? Please extend your arguments
Response: Line 39-45 on page 1. Thanks for the comments. We accepted the suggestions, and added relevant content.
According to the bulletin on ecological and environmental conditions of China, more than 11,800 natural protected areas had already established by 2019 in China, covering a total area of about 2 million km2, accounting for 18.0 percent of the country’s land area and 5.80 percent of the global protected areas. China is in the critical period of transition from focusing on the quantity of protected areas to paying attention to the role of protected areas. Therefore, it is of practical significance to select China as the research sample to study the ability of protected areas to relieve the pressure of human activities.
- Lines 47-48: The international relevance of this contribution should be highlighted and mentioned. Also, a research question should be formulated at this section.
Response: Line 45-59 on page 1. Thanks for the comments. We accepted the suggestions, and revised it.
The core question of this study is to what extent do protected areas in China mitigate the pressures of human activities, and what will happen in the future? This paper has three aspects of innovation. In this study, a human activity pressure (HAP) index is constructed as an evaluation index. On the basis of solving the problem of sample selectivity bias by propensity score matching, this paper evaluates the effectiveness of China's PAs in alleviating HAP. We also explore the impact of the pressure of a certain human activity on PAs in future scenario. Firstly, this paper constructs the HAP from 2000 to 2020 coupled with multi-source data. Secondly, after the propensity score, matching method was used to eliminate the sample selectivity bias, the relative effectiveness index and panel model are used to evaluate the effectiveness of China's PAs in alleviating the pressure of human activities from different levels. Finally, the propensity score matching method is used to evaluate the effectiveness of China's PAs in alleviating population pressure and urban land pressure under the SSPs scenario. This research results can provide scientific basis for global protected areas to relieve the pressure of human activities.
- Also, hypothesis mentioned in method section should be inserted in introduction section.
Response: Line 183-185 on page 4. Thanks for the comments. The hypothesis in the method section is about how to deal with the variable of population pressure. It is the process of data processing, and not the core hypothesis of the whole paper. Therefore, we decide not to put it in the introduction section. The content of the original text is as follows:
“The modeling of population pressure adopts the formula method. It is assumed that the population pressure on the natural system increases logarithmically with the increase of population density and can reach saturation after a certain point, that is to say, the population pressure remains unchanged after the population density increases to a certain degree. For this hypothesis, when the population density is greater than 1000 people /km2, the score of population pressure is 10, when the population density is lower than 1000 people /km2, the score of population pressure is calculated by formula”
6.Lines 166-167: Why this kind of HAP are selected? This should be explained more in depth in this paragraph.
Response: Line 168-172 on page 4. Thanks for the comments. We accepted the suggestions, and added relevant content.
Human activity pressure data products are divided into single pressure products and comprehensive pressure products. At present, comprehensive pressure evaluation is the mainstream evaluation method, among which Venter et al. [19] constructed the human footprint method, which is the most representative, and estimated the global human activity pressure. Therefore, this paper selects four kinds of single HAP, namely urban land, farmland, population and power facilities, to construct a comprehensive HAP index.
- Figures 1 and 2 should be improved in terms of quality.
Response: pages 7 and 8. Thanks for the comments. We improved Figures 1 and 2 in terms of quality. We adjusted the resolution of the image to 600dpi. In addition, we adjusted other details of the image, such as keeping the decimal point to two digits.
- Conclusions should discussion about the potential implications of result to the international context, especially in terms of validity of the method applied. A new paragraph could be inserted in line 424, or in line 432.
Response: Line 428-430 on page 15. Thanks for the comments. We accepted the suggestions, and added relevant content.
“Secondly, in studies evaluating the effectiveness of PAs, this paper presents a new potential approach, the pressure of human activities is used as an indicator [35, 44] or the state of natural ecosystems as an indicator [16, 22, 32]. But the interaction between human activities and nature is a complex process, and unilateral assessment has certain limitations. Future research should focus on the interaction process between human activities and nature, so as to provide more reasonable guidance for the planning, construction, evaluation and management of PAs.”

Reviewer 2 Report
The paper claims that few research use human activity pressure to assess the effectiveness of protected areas in China. The study proposes a human activity pressure index and uses propensity score matching method as well as panel model to address the research problem in relation to assessing the effectiveness of China's protected areas at mitigating human activity pressure. It also predicts the change trend in 2050 under the SSPs scenario. A wide range of recent and relevant academic literature was listed in the References but there's no engagement at with recent debates in the International Journal of Environmental Research and Public Health. The Authors need to situate their research within relevant discussions within this journal and cite a few publications from this journal to justify publication in this journal.
Author Response
Response: Line 102-115 on pages 4 and 5. Thanks for the comments. We added articles related to protected areas research in this research journal, and discussed and responded to their research.
- Line26-27 on page 1
Although the coverage of PAs is increasing, it is not completely immune to the impact of human activities[1, 2].
- Romanillos, T.; Maneja, R.; Varga, D.; Badiella, L.; Boada, M., Protected Natural Areas: In Sickness and in Health. Int J Env Res Pub He 2018, 15, (10), 2182.
(2) Line161-165 on page 4
In their research, several matching methods were compared, and finally, propensity score matching (PSM) was chosen to eliminate the sample selectivity bias, and the effectiveness of alleviating the pressure of human activities in the PAs was reasonably evaluated, which proved the scientific nature of this evaluation method[3-5].
- Zhang, Z.; Tang, Y.; Pan, H.; Yao, C.; Zhang, T., Assessment of the Ecological Protection Effectiveness of Protected Areas Using Propensity Score Matching: A Case Study in Sichuan, China. Int J Env Res Pub He 2022, 19, (8), 4920.
(3) Line383-385 on page 14
In addition, conservation of PAs is not uniform across China's provinces, which may also contribute to the differences in the relief of HAP in PAs[6].
- Jiricka-Pürrer, A.; Tadini, V.; Salak, B.; Taczanowska, K.; Tucki, A.; Senes, G., Do Protected Areas Contribute to Health and Well-Being? A Cross-Cultural Comparison. Int J Env Res Pub He 2019, 16, (7), 1172.
